# Validation and Trend Analysis of Stratospheric Ozone Data from Ground-Based Observations at Lauder, New Zealand

**Leonie Bernet** [1,2,*] **, Ian Boyd** [3] **, Gerald Nedoluha** [4] **, Richard Querel** [5] **, Daan Swart** [6] **and Klemens Hocke** [1,2]

1 Institute of Applied Physics, University of Bern, 3012 Bern, Switzerland; klemens.hocke@iap.unibe.ch
2 Oeschger Centre for Climate Change Research, University of Bern, 3012 Bern, Switzerland
3 BC Scientific Consulting LLC, Stony Brook, NY 11790, USA; iboyd@astro.umass.edu
4 Naval Research Laboratory, Remote Sensing Division, Washington, DC 20375, USA; gerald.nedoluha@nrl.navy.mil
5 National Institute of Water & Atmospheric Research, Lauder 9377, New Zealand; richard.querel@niwa.co.nz
6 National Institute for Public Health and the Environment (RIVM), 3720 Bilthoven, The Netherlands; daan.swart@rivm.nl
* Correspondence: leonie.bernet@iap.unibe.ch

**Abstract:** Changes in stratospheric ozone have to be assessed continuously to evaluate the effectiveness of the Montreal Protocol. In the southern hemisphere, few ground-based observational datasets exist, making measurements at the Network for the Detection of Atmospheric Composition Change (NDACC) station at Lauder, New Zealand invaluable. Investigating these datasets in detail is essential to derive realistic ozone trends. We compared lidar data and microwave radiometer data with collocated Aura Microwave Limb sounder (MLS) satellite data and ERA5 reanalysis data. The detailed comparison makes it possible to assess inhomogeneities in the data. We find good agreement between the datasets but also some possible biases, especially in the ERA5 data. The data uncertainties and the inhomogeneities were then considered when deriving trends. Using two regression models from the Long-term Ozone Trends and Uncertainties in the Stratosphere (LOTUS) project and from the Karlsruhe Institute of Technology (KIT), we estimated resulting ozone trends. Further, we assessed how trends are affected by data uncertainties and inhomogeneities. We find positive ozone trends throughout the stratosphere between 0% and 5% per decade and show that considering data uncertainties and inhomogeneities in the regression affects the resulting trends.

**Keywords:** stratospheric ozone; trends; ozone profiles; microwave radiometry; lidars

## 1. Introduction

Stratospheric ozone protects life on earth from harmful solar UV radiation and is involved in multiple radiative, chemical, and dynamic processes (e.g., [1–7]). Changes in stratospheric ozone have to be assessed continuously to verify how it reacts to the decline of ozone-depleting substances (ODSs) and to a changing climate. Anthropogenic ODS emissions caused a strong decrease in stratospheric ozone observed from the 1960s. The Montreal Protocol (1987) succeeded in reducing ODS emissions. Consequently, concentrations of stratospheric chlorine have been decreasing since 1997 [8]. Recent studies report that stratospheric ozone over Antarctica is responding to these changes and starting to recover [9–15]. Outside of the polar regions, however, differences in magnitude and uncertainty of ozone trends are reported [16–18]. Even though consensus exists that stratospheric ozone has stopped declining in the late 1990s [7,19–24], a general increase in stratospheric ozone has proved difficult to detect, and positive ozone trends at midlatitudes have recently been recorded only in the upper stratosphere (e.g., [7,17,18]).

Stable long-term measurements of stratospheric ozone are indispensable to assessing changes in stratospheric ozone. Ground-based measurements are crucial not only to validate satellite data but also because they provide stable measurements with few instrumental changes over many decades. In the southern hemisphere (SH) midlatitudes,

continuous stratospheric ozone measurements are rare. Ozone observations at Lauder, New Zealand are therefore invaluable to derive stratospheric ozone trends at southern midlatitudes. The Lauder ozone measurements are part of the Network for the Detection of Atmospheric Composition Change [25] and have provided stable, continuous measurements since the early 1980s. Recently, Lauder stratospheric ozone data have been used in several trend studies and reports, such as Steinbrecht et al's [17] study and the comprehensive report on Long-term Ozone Trends and Uncertainties in the Stratosphere (LOTUS) [18]. However, the collocated time series of various ground-based instruments at Lauder have not been compared in detail. Investigating the time series in detail is required to assess their suitability for trend estimations.

The aim of the present study is to investigate ozone time series at Lauder and to verify whether they are suitable for trend estimation. For this purpose, we compare coincident measurements of microwave radiometer, lidar, ERA5 reanalysis data, and satellite data from the Aura Microwave Limb Sounder (Aura MLS) at Lauder to identify possible inhomogeneities. If one dataset deviates from the others, it might be the result of measurement biases, which are then considered in the trend estimation. Further, we compare two trend analysis methods, the LOTUS regression [18] and the KIT model developed at the Karlsruhe Institute of Technology (KIT) [26]. Both models are based on multiple linear regression, using slightly different regression parameters. Further, they handle data uncertainties in a different way: The KIT model considers data uncertainties, whereas these have not been considered in the LOTUS model for ground-based data so far [18]. Finally, we assess how inhomogeneities in the time series and data uncertainties affect the trend estimates.

## 2. Ozone Datasets

We use stratospheric ozone data (1997 to 2019) from a microwave radiometer (MWR) and a lidar, both located at Lauder, New Zealand (45°S, 169.7°E, 370 m above sea level (a.s.l.)). Both instruments are part of NDACC, and the data are archived on the NDACC website [25]. In addition, we use ozone profiles from the MLS on the Aura satellite [27] and ERA5 reanalysis data [28]. We limit our analyses to altitudes from 14 to 50 km and refer for convenience to the lower stratosphere (14 to 20 km), the middle stratosphere (20 to 30 km), the upper-middle stratosphere (30 to 39 km), and the upper stratosphere (39 to 50 km).

### 2.1. Microwave Radiometer

The Microwave Ozone Profiling Instrument (MOPI) is an MWR that measured stratospheric ozone at Lauder from 1992 to 2016. It measures ozone emission of ozone molecules due to rotational transitions at 110.836 GHz at a 20-minute resolution. The measured spectra are then used to retrieve ozone volume mixing ratio (VMR) profiles from 20 to 68 km, with a vertical resolution of around 7 to 8 km at 10 hPa [29]. Measurements are performed in clear-sky and some overcast conditions and are averaged to obtain up to two daytime and two nighttime retrievals per day [29]. We use the most recently retrieved MWR data (version 6) that are available on NDACC [25] (last access: 1 April 2020) and at https://doi.org/10.21336/gen.bpqv-7z42 [30]. The MOPI instrument and data have been described by Nedoluha et al. [29], and basic technical details about the measurements and the instrument are given in Parrish et al.'s [31] and Parrish's [32] studies. The data have been validated by Boyd et al. [33], showing a general agreement within 5% with Aura MLS data.

### 2.2. Lidar

The stratospheric ozone lidar at Lauder is a differential absorption lidar (DIAL) that has been measuring since November 1994. It emits laser pulses at wavelengths of 308 and 353 nm, of which the first is strongly absorbed by ozone molecules. The ratio of the backscattered signal at both wavelengths and the signal travel time provides information about the vertical ozone distribution. The system has been described in Swart et al. [34]. Very good agreement with ozonesonde and satellite data has been shown

by Keckhut et al. [35]. Further, the lidar data have been validated within multiple NDACC intercomparison studies (e.g., [36–38]). We use the lidar data available on NDACC (2020) (processing version 8.3, last access: 31 March 2020) and at https://doi.org/10.21336/gen. 0x48-sm13 [39]. The lidar retrieves ozone during clear sky nights, with an average of five profiles per month within our study period. The sparse sampling might result in the distortion of monthly means. For trend estimation, we therefore applied a seasonal fitting on daily lidar means. We then show trend estimates of both monthly lidar time series. We use the term "lidar fit" when referring to the lidar data with seasonal fit, whereas "full lidar" refers to regular monthly lidar means. For the seasonal fit, a seasonal model is fit to the 15th of each month at each altitude level using a specific window length, following Wilhelm et al.'s [40] method as described in Bernet et al. [41]. Instead of their proposed window length of 2 years, we used a length of 1.5 years. We judged this to be sufficient for the lidar data, which is generally well distributed within a month. Monthly means were excluded wherever less than 50 measurements were available in the window.

Lidar ozone profiles were initially given in number density and converted with coincident ERA5 pressure and temperature profiles to VMR. We limited the lidar data to altitudes from 14.2 to 38.6 km, where the averaged measurement uncertainty within our study period remains below 5%. To compare the vertically highly resolved lidar profiles with less resolved MWR profiles, the lidar data were convolved with the MWR averaging kernels according to Connor et al. [42]. To do this, the rows of the averaging kernels were interpolated to the highly resolved lidar grid and scaled to conserve the measurement response [43].

### 2.3. Aura MLS

The Microwave Limb Sounder (MLS) on board the Aura satellite has been providing profiles of stratospheric ozone since August 2004 [44]. It retrieves ozone from radiance measurements at 240 GHz. The ozone data have been validated by Froidevaux et al. [45]. We use Aura MLS data version 4.2. [27], which is described in detail by Livesey et al. [46]. The satellite crosses Lauder twice a day at a spatial coincidence of $\pm 1^\circ$ latitude and $\pm 8^\circ$ longitude.

### 2.4. ERA5

ERA5 is the atmospheric reanalysis from the European Centre for Medium Range Weather Forecasts (ECMWF) [47]. We derived six hourly ozone VMR profiles on model levels from ERA5-complete ozone mass mixing ratio profiles provided by Copernicus Climate Change Service (C3S) [28]. The reanalysis model assimilates various ozone satellite measurements to derive ozone profiles, as described by Hersbach et al. [47]. When comparing ERA5 profiles with MWR profiles, the ERA5 profiles were convolved with MWR averaging kernels as described above.

## 3. Time Series Comparison

Deseasonalized anomalies of MWR, lidar, and ERA5 data show that stratospheric ozone at Lauder varies naturally within around 10% in the middle and upper stratosphere (Figure 1a–c). The anomalies are defined as the difference between a monthly mean and the overall mean value of each month, illustrating the interannual variability of the data. Occasionally, larger anomalies are observed, especially in the lower stratosphere (Figure 1d). All datasets agree on specific natural anomalies. This includes, for example, increased ozone between 30 and 40 km from 2009 to 2013, as also reported by Nedoluha et al. [29], an ozone minimum in November 1997 [48], and an ozone minimum in 2007 between 20 and 30 km. Aura MLS and ERA5 agree closely, which is expected because Aura MLS data is assimilated in ERA5. However, we also observe some differences between the datasets. For example, ERA5 deviates from MWR in the middle stratosphere from 2011 to 2014, and the ERA5 deviates from the lidar after 2015 in the lower and middle stratosphere.

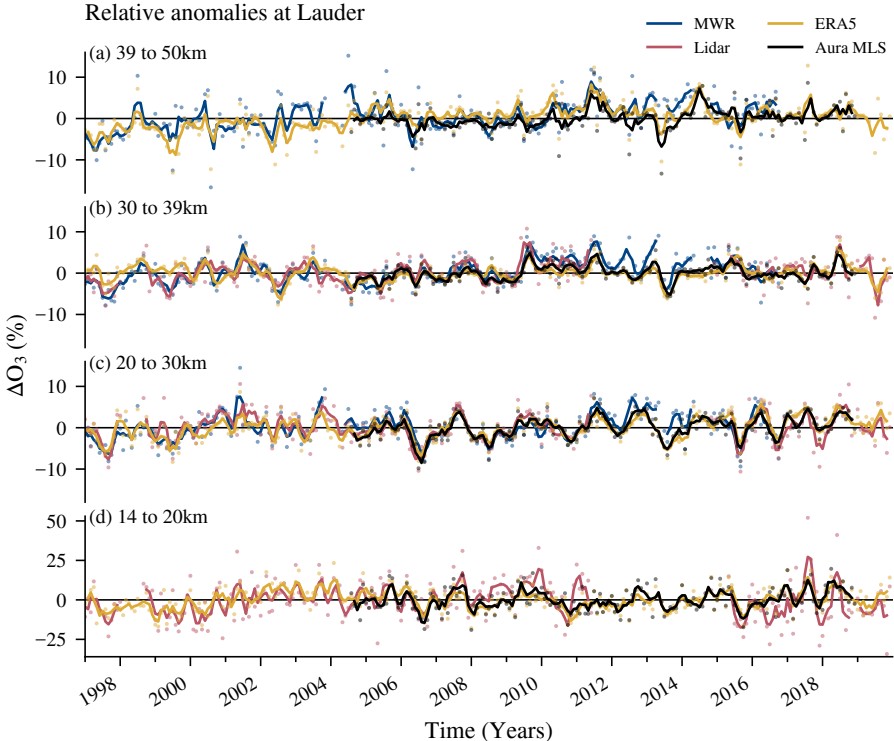

**Figure 1.** Relative anomalies of monthly ozone volume mixing ratio (VMR) at Lauder, New Zealand, for microwave radiometer (MWR), lidar, ERA5 reanalysis, and Aura MLS data. The relative anomalies (dots) show for each month the deviation from the monthly mean climatology (1997 to 2019). The bold lines show data smoothed with a moving window of three months. Relative anomalies are averaged over four altitude ranges, representing approximately (**a**) the upper stratosphere, (**b**) the upper-middle stratosphere, (**c**) the middle stratosphere, and (**d**) the lower stratosphere.

To better evaluate such differences, we compared monthly means of MWR and lidar profiles with coincident ERA5 profiles with a time coincidence of ±3 h. In addition, we compared monthly means of the two ground-based instruments (MWR and lidar), with a time coincidence of ±6 h. These relative differences are shown in Figure 2. The MWR and lidar data mostly agree well, with slightly more MWR ozone in the middle stratosphere and more ozone observed by the lidar in the upper-middle stratosphere to upper stratosphere (Figure 2a). However, ERA5 strongly underestimates ozone in the upper stratosphere compared to MWR (Figure 2b). Further, ERA5 reports slightly larger ozone values than MWR in the upper-middle stratosphere, except from 2009 to 2014. The same pattern is also observed when comparing ERA5 with lidar data (Figure 2c): ERA5 underestimates ozone in the upper stratosphere and overestimates ozone in the upper-middle stratosphere. For lidar data, we further observe that the difference compared to ERA5 increases after the data gap in 2014 in the lower stratosphere. This might be due to potential lidar changes after the data gap or due to increased ERA5 anomalies after 2015, which are also reported by Hersbach et al. [47]. We therefore consider this change for both lidar and ERA5 data when determining trends with the KIT trend model (see Section 4.1.2). Changes in lower stratospheric lidar data in 2018 might be related to the addition of two low-altitude channels in the lidar retrieval in October 2018. Further, a change in lidar is observed compared to ERA5 in 2004, with better agreement in the middle stratosphere after 2004. This change is not observed in the comparison between lidar and MWR data (Figure 2a), which suggests that it is due to a change in ERA5 data. This confirms ERA5 anomalies observed by Hersbach et al. [47], who attribute the change to the assimilation of Aura MLS measurements in ERA5 ozone data from that time. We later account also for this change when determining ERA5 trends, as described in Section 4.1.2.

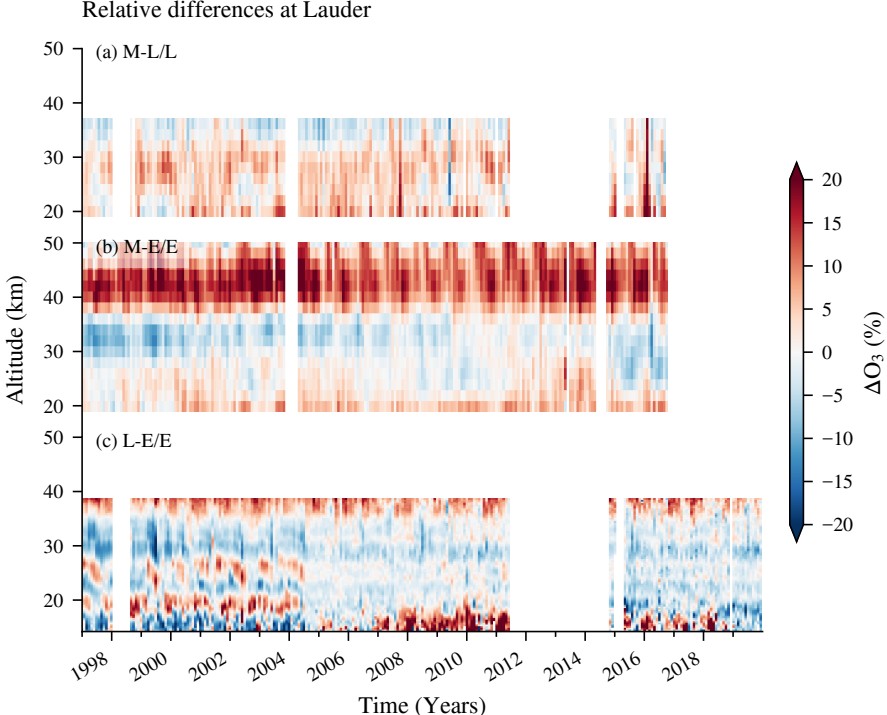

**Figure 2.** Relative ozone differences between (**a**) MWR (M) and lidar (L) data, (**b**) MWR and ERA5 (E) data, and (**c**) lidar and ERA5 data. Lidar and ERA5 profiles in (**a,b**) have been convolved with MWR averaging kernels.

## 4. Trend Estimations

### 4.1. Trend Models

We use two multiple linear regression models and compare the trend estimates using both methods. The first model is the LOTUS regression model (available at https://arg.usask.ca/docs/LOTUS_regression/), which was developed within the LOTUS activity on stratospheric ozone trends and uncertainties and is described in detail in SPARC/IO3C/GAW [18]. The second model is a multiple linear trend model developed at the Karlsruhe Institute of Technology (KIT), described by von Clarmann et al. [26]. Both models fit monthly ozone data with a multiple linear regression function by minimizing a cost function. They account for autocorrelation between residuals within an iterative process and use the following predictors: the Quasi biannual oscillation (QBO), the El Niño Southern Oscillation (ENSO), solar activity as well as four periodic oscillations to account for seasonality [18,49]. The KIT model uses normalized Singapore winds at 30 and 50 hPa for QBO (available at https://www.geo.fu-berlin.de/met/ag/strat/produkte/qbo/singapore.dat); the multivariate ENSO Index (MEI, available at https://psl.noaa.gov/enso/mei/data/meiv2.data); and solar flux data measured at a wavelength of 10.7 cm [50]. The LOTUS model uses two orthogonal components of the QBO derived with a principal component analysis from Singapore wind data (from https://www.geo.fu-berlin.de/met/ag/strat/produkte/qbo/qbo.dat); the same MEI data as the KIT model; and solar flux data from https://omniweb.gsfc.nasa.gov/form/dx1.html. The LOTUS model further uses two additional predictors: tropopause pressure (from ftp://ftp.cdc.noaa.gov/Datasets/ncep.reanalysis.derived/tropopause/pres.tropp.mon.mean.nc) and aerosol optical depth (AOD). The AOD data is used from the Goddard Institute for Space Studies (https://data.giss.nasa.gov/modelforce/strataer/) and extended after 2012 by extrapolating the last value [18]. The AOD, however, should be negligible because no significant volcanic erruption occurred in our study period. For the LOTUS regression, we use a piecewise linear term (PWLT) as predictor, with a linear increase starting in 1997 [18].

We determine linear trends by fitting the regression functions to monthly ozone data. We start the trend estimates in 1997, when a turnaround due to decreasing ODSs is expected [8]. MWR data is not available after 2016, so we limit the trend estimates to that year to compare trends in all datasets over the same period. Trends are considered to be significantly different from zero at 95% confidence intervals when they exceed twice their standard deviation.

### 4.1.1. Weighted Regression

The KIT model uses the uncertainties of the monthly means to weight the regression [26,49]. The weighting is performed by using a full error covariance matrix in the cost function, where the diagonal elements represent the monthly uncertainties. This improves the regression fit and results in more realistic trend uncertainties. The feature of weighted regression is also available in the LOTUS model but has not been used for final trend results in SPARC/IO3C/GAW [18] due to the difficulty of correcting for unknown variances in the data (heteroscedasticity correction). This is mainly problematic when using merged datasets in which the sampling frequency and thus the monthly standard errors change over time [18]. In our case, the sampling frequency is rather constant over time, and we therefore apply the weighted LOTUS regression to the ozone time series to compare it with the unweighted regression.

To weight the regression, the diagonal elements of the covariance matrix are set to the monthly uncertainties of the data. We estimate monthly uncertainties for MWR data by

$$\sigma_{MWR} = \sqrt{\sigma_{rand}^2 + \sigma_{sys}^2 + \sigma_{\bar{x}}^2}, \tag{1}$$

where $\sigma_{rand}$ is the random measurement error, $\sigma_{sys}$ is a systematic error, and $\sigma_{\bar{x}}$ is the standard error of the monthly mean, given by

$$\sigma_{\bar{x}} = \sigma\, n^{-\frac{1}{2}}, \tag{2}$$

where $\sigma$ is the standard deviation of the monthly mean and $n$ the number of measurements per month. The lidar monthly uncertainties are given by

$$\sigma_{lidar} = \sqrt{\sigma_{sys}^2 + \sigma_{\bar{x}}^2}, \tag{3}$$

where $\sigma_{sys}$ is the systematic measurement uncertainty from photon-counting statistics. For the fitted lidar data, an additional error term resulting from the seasonal fit is included:

$$\sigma_{lidarfit} = \sqrt{\sigma_{sys}^2 + \sigma_{\bar{x}}^2 + \sigma_{fit}^2}. \tag{4}$$

For ERA5 data, the standard error is small due to the high temporal resolution. To our knowledge, no comprehensive ozone cross-comparisons have yet been published of the new ERA5 data that derive systematic uncertainties. We therefore add a systematic uncertainty of 5% to the standard error of each ERA5 monthly mean. This corresponds approximately to the averaged difference between ERA5 and MWR profiles in our study period.

### 4.1.2. Bias Correction

An additional feature of the KIT model is the possibility of accounting for biases within the trend estimation. This is helpful if the data shows some jumps or inhomogeneities, for example after instrumental changes. We therefore apply this approach to account for the jumps that we identified in the lidar and ERA5 data (Section 3) when estimating trends. For this purpose, a fully correlated block is added to the error covariance matrix in the weighted regression. The block, corresponding to the biased subset, is filled with the square of the estimated bias uncertainty. This enables a fit of the bias and is mathematically equivalent to adding the bias as an independent fit variable [51]. The bias can thus be estimated from

the data itself and does not depend on an a priori choice of bias [52]. The bias uncertainty chosen determines how much freedom the programme has when estimating the bias from the data. Following Bernet et al.'s [49] suggestion, we use an altitude-independent bias uncertainty of 5 ppm. They assessed this value to be appropriate for fitting the bias from the data independently of the a priori zero bias.

In the present study, we apply this bias correction to lidar and ERA5 data, for which we identified inhomogeneities as described in Section 3. To this end, we identified change points in the data based on the inspection of Figure 2. We considered a change point in the lidar data after the instrumental break in 2014, when we identified anomalies in the lower stratosphere compared to previous data. For ERA5, we consider change points in 2004 and in 2015 that have been identified by Hersbach et al. [47] and confirmed in our comparison with the lidar data (see Section 3). The data block after the change point is then assumed to be biased compared to the previous data, and a bias is fitted from the data in the subsequent block. The programme estimates a bias for the biased block at each altitude, leading to the bias profiles shown in Figure 3. Note that to obtain the corrected time series, these bias profiles have to be subtracted from the original time series in the biased subsets.

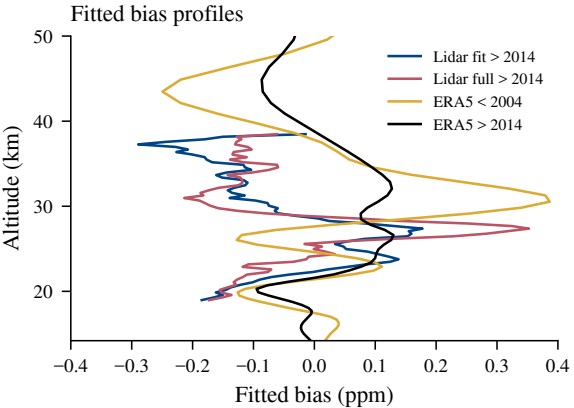

**Figure 3.** Bias profiles as estimated by the KIT regression for lidar and ERA5 data. The lidar bias compares to data before 2014, whereas the data block from 2004 to 2015 is the reference for ERA5 biases.

### 4.1.3. Artificial Test Case

To illustrate weighted regression, we present an artificial test case in Figure 4. The corresponding estimated trends for four different scenarios using the KIT and the LOTUS model are given in Table 1. The artificial time series has a trend of 0.1 ppm per decade. We added anomalies to the summer months of 2012, 2014, and 2015. Such anomalies could represent for example months with few measurements and larger uncertainties. The trend of this biased time series is then overestimated (case B), with a trend estimate of 0.13 ppm (KIT) or 0.14 ppm (LOTUS) per decade instead of the true trend of 0.1 ppm per decade (Table 1). We therefore adapted the uncertainties for these months to weight the regression, as shown in Figure 4b. Considering the adapted uncertainties in the covariance matrix changes the trend fit for both models (case C), which is then closer to the true trend (Table 1). The weighted LOTUS trend (case C) is slightly overcorrected, suggesting that further investigations might be necessary to use the LOTUS weighting with confidence. When the bias fit in the KIT model is applied (case D), the trend corresponds to the true trend, which has also been demonstrated by Bernet et al. [49].

Further, we found that the weighting is less effective if a data jump with a subsequent biased block is added to the artificial time series (not shown). In such a case, the KIT bias fitting corrects the trend estimate, as shown by Bernet et al. [41]. The simple weighting, however, is not sufficient to correct for such a jump. Moreover, we found that the LOTUS model would require additional adjustments to estimate trends with a data jump, including

for example a heteroscedasticity correction to account for the varying residuals, as described by Damadeo et al. [53] and SPARC/IO3C/GAW [18]. Further investigations would be necessary to derive solid conclusions about such corrections in the LOTUS model.

We conclude that using weighted regression changes trend fits in both regression models. Depending upon the model being used, the trend may differ by 0.06 ppm per decade in this case.

**Table 1.** Trends for the artificial time series with different corrections as shown in Figure 4, using the KIT and the LOTUS trend model.

| Case | Characteristics | KIT Trends (ppm decade$^{-1}$) | LOTUS Trends (ppm decade$^{-1}$) |
|------|-----------------|--------------------------------|----------------------------------|
| (A) | True | $0.10 \pm 0.07$ | $0.10 \pm 0.00$ |
| (B) | Biased | $0.13 \pm 0.07$ | $0.14 \pm 0.03$ |
| (C) | Biased and weighted | $0.11 \pm 0.07$ | $0.08 \pm 0.03$ |
| (D) | Bias-corrected | $0.10 \pm 0.12$ | – |

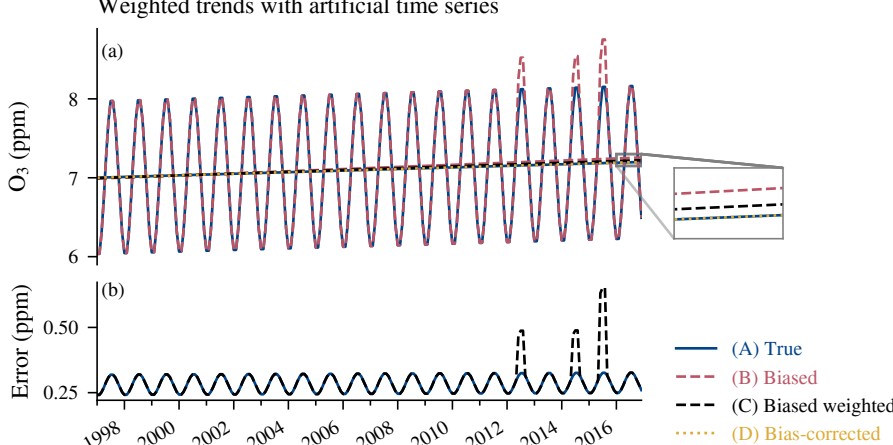

**Figure 4.** (**a**) Artificial time series with a trend of 0.1 ppm per decade (case A) and with added anomalies in 2012, 2014, and 2015 (case B). (**b**) Uncertainties used for the trend estimation, with increased uncertainties in 2012, 2014, and 2015 when using the weighted trend. The straight lines shown in the magnified rectangle in panel (**a**) show the estimated linear trends derived with the KIT model, for the true time series (case A), for the time series with added anomalies but weighted with the regular uncertainties (case B), and for the time series with added anomalies weighted with adapted uncertainties (case C). In case D, a bias was fitted within the KIT model for the anomalous periods. Case D agrees best with the true trend (case A).

### 4.2. Ozone Trend Estimates

Trend profiles have been estimated using the LOTUS regression (Figure 5) and the KIT regression model (Figure 6). Both trend models report generally positive ozone trends between 0% and 5% per decade in the middle and upper stratosphere. Only ERA5 shows a negative trend peak at 30 km. This seems to be an artefact that is probably related to the start of Aura MLS assimilation in 2004, leading to a jump in ERA5 data as shown in Figure 2c. MWR trends are almost zero between 25 and 30 km, and peak in the upper stratosphere. Lidar trends vary between 0% and 3% per decade in the middle stratosphere. They are negative (lidar full) or close to zero (lidar fit) at 20 km and increase strongly in the lower stratosphere. Differences between the full lidar trends (lidar full) and trends using lidar data with the seasonal fit (lidar fit) are small in the middle stratosphere. In the lower stratosphere, however, the lidar trend with seasonal fit (lidar fit) is unrealistic high, indicating that the seasonal fit may not be able to handle the large variability at these altitudes.

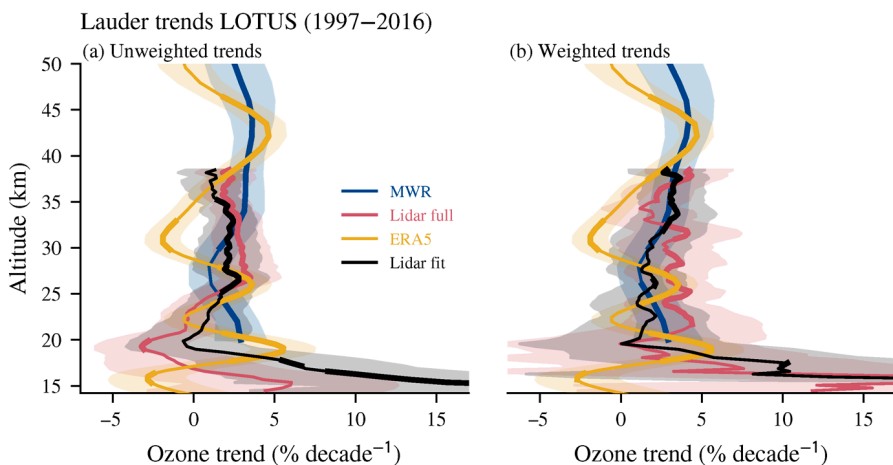

**Figure 5.** Ozone trends at Lauder from January 1997 to December 2016 for MWR, full lidar data (lidar full), ERA5 data, and lidar data where a seasonal fit was applied (lidar fit). The trends have been estimated using the LOTUS regression model. Panel (**a**) shows the unweighted trend estimates, whereas monthly means have been weighted by their uncertainties in the regression fit in (**b**). Shaded areas represent 2-standard-deviation ($\sigma$) uncertainties, and bold lines mark trends that are significantly different from zero at 95% confidence intervals ($|\text{trend}| > 2\sigma$).

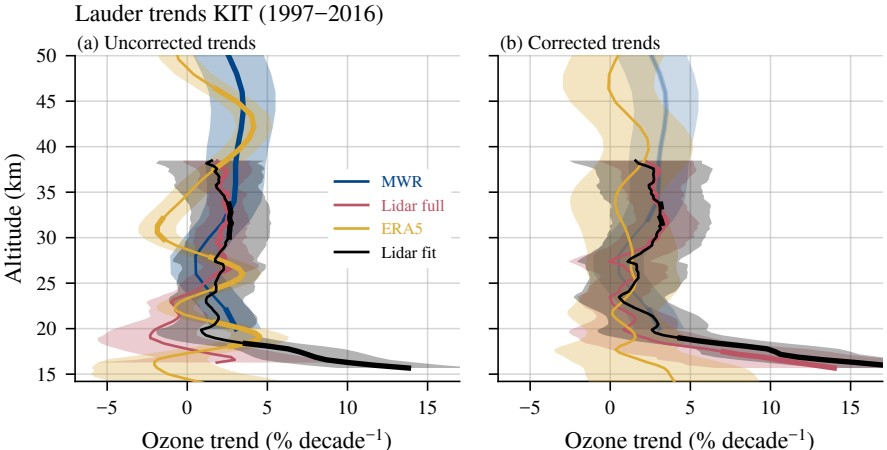

**Figure 6.** Ozone trends estimated with the KIT regression model for MWR, full lidar, ERA5, and seasonal lidar data (1997 to 2016). Trends are weighted but uncorrected with respect to biases in panel (**a**). Panel (**b**) shows weighted bias corrected trends for lidar and ERA5. Uncorrected MWR trends are also shown in (**b**) (pale color) for comparison.

When using weighted LOTUS regression (Figure 5b), MWR and fitted lidar trends are consistent in the middle stratosphere. They agree better than the unweighted trends (Figure 5a), suggesting that the weighting improves the lidar trend estimate. Further, the negative lidar trend peak when using full lidar data at 20 km is not visible in the weighted trend estimate. The weighted trends for full lidar data are noisier throughout the stratosphere than the fitted lidar data. This might be due to the larger variability of the full lidar monthly means and uncertainties compared to the fitted lidar data. The seasonality of the uncertainties might also affect the weighted trends. In the lower stratosphere, the weighted lidar trends are high at the lowest altitude levels. These trends are probably overestimated when expressed in percent due to the small amount of ozone volume mixing ratio in the lower stratosphere. A small trend difference at these altitudes may be overweighted when expressed in percent, whereas the same trend difference would be less visible in the middle stratosphere where the ozone volume mixing ratio is larger. This effect may be further amplified by the large uncertainties that we use at these altitudes due to

increased measurement uncertainties and the large ozone variability. Finally, weighted ERA5 and MWR LOTUS trends do not differ much from the unweighted LOTUS trends.

The KIT trend profiles (Figure 6a) are similar to the LOTUS trends (Figure 5). However, small differences in the middle stratospheric lidar trends exist, which might be related to different model setups including differences in predictors. Further, the KIT full lidar trend profile is less variable than the weighted LOTUS profile which suggests differences in the weighting procedures for uncertainties which are more variable in time.

To account for inhomogeneities in the time series, the lidar and ERA5 trend profiles were corrected for observed anomalies as described in Section 4.1.2. The bias-corrected lidar and ERA5 trend profiles are shown in Figure 6b. The corrected lidar profiles differ from the uncorrected trend profiles (Figure 6a) mainly in the lower stratosphere, with a better agreement to the MWR profile between approximately 20 and 25 km. This corresponds to the altitudes where the bias was observed (Figure 2c), suggesting that the bias was successfully considered in the corrected trend estimate. Below 20 km, the lidar trends are unrealistic high, which may be related to natural variability, the large instrumental uncertainties used at these altitudes, and a possible overweighting when expressing trends in percent, as described above. In the corrected ERA5 trend profile, the negative ERA5 trends at 30 km and in the lower stratosphere are reduced, and the corrected ERA5 profile agrees more closely with the MWR and lidar trend profiles.

## 5. Discussion of Results

We have shown that the Lauder ozone datasets agree remarkably well on ozone anomalies from 1997 to 2019. Further, the two ground-based ozone datasets at Lauder agree well, with differences mostly below 10%. The good agreement proves the quality of both ground-based datasets. Differences from ERA5 are larger, especially in the upper stratosphere, where ERA5 strongly underestimates ozone compared to MWR and lidar data. By comparing the various Lauder datasets, we identified data-specific inhomogeneities, especially in ERA5 data. Such data inhomogeneities in the time series may impact trend estimates and their uncertainties. We observe inhomogeneities in ERA5 data in 2004 and 2015. For the lidar, we observe small changes in the data after pausing measurements in 2014. Whereas the lidar and MWR can be considered suitable for trend estimations, the ERA5 data requires corrections for biases when estimating trends. Indeed, trends from reanalysis data should generally be handled with care because of unconsidered changes in observing systems of assimilated data (e.g., [54]). To improve the lidar trend, we also considered the change observed after the instrumental break in 2014 by fitting a bias to the anomalous period using the trend model from von Clarmann et al. [26].

We have presented two regression models and determined trends using unweighted, weighted, and bias-corrected regression. Unweighted lidar and MWR trends agree well in the middle stratosphere and differ in the lower stratosphere, whereas unweighted ERA5 trends disagree in the middle stratosphere, apparently as the result of biases in the data. In most stratospheric ozone trend studies, data uncertainties are not considered when estimating trends [18]. However, considering the uncertainties of the time series can improve the trend estimates and their uncertainties. We therefore use weighted regression with the KIT model and apply weighted regression within the LOTUS trend model to account for the time dependence of data uncertainties. Logically, the weighted regression should be more reliable than the unweighted regression, as also shown with our artificial time series. However, the weighted LOTUS full lidar trends show larger variability with altitude, suggesting that additional model adjustments might be required when weighting with varying uncertainties. Further investigations might be necessary to derive sound conclusions about the use of the LOTUS model with the option of weighted regression. For example, a heteroscedasticity correction, as suggested by SPARC/IO3C/GAW [18] and presented by Damadeo et al. [53], might improve the weighted trend estimates. Bias-corrected trends are presented using the KIT regression. For this, we corrected lidar and ERA5 trends by fitting a bias to anomalous periods. This bias correction affects the lidar trend estimate, which then

agrees more closely with the MWR trend profile in the lower-middle stratosphere. Moreover, the bias-corrected ERA5 trend profile agrees more closely with the MWR and lidar trends than the uncorrected profile. Our trend results generally show that weighted and corrected trend estimates change the trend values and their uncertainties slightly. Nevertheless, the changes generally lie within the trend uncertainties.

Our lidar and MWR trends in the middle and upper stratosphere agree on values between 2% and 3% per decade. Further, all our datasets report positive stratospheric ozone trends, with the exception of a negative trend peak reported by the uncorrected ERA5 trend profile at around 30 km and a negative trend when using full lidar data at 20 km. However, this negative lidar trend is not visible in the weighted LOTUS and bias-corrected KIT trend, suggesting that it is caused by data inhomogeneities. Depending on the regression model used, trends are significantly positive in the middle and the upper stratosphere. Our trends in the middle and upper stratosphere are consistent with other studies. In the upper stratosphere, significant positive trends were reported at Lauder from Fourier transform infrared (FTIR) observations (2001 to 2012, Vigouroux et al. [55]). Similar trends were also reported from combined ground- and space-based data at Lauder by Nair et al. [56] (1997 to 2012) and from various Lauder instruments presented in SPARC/IO3C/GAW [18]. In the lower stratosphere, negative to near-zero trends are reported by SPARC/IO3C/GAW [18] for SH midlatitudes using a range of satellite records. Further, ozonesonde and FTIR observations at Lauder [18] indicate negative lower-stratospheric trends, which are also reported by Zerefos et al. [57] based on Solar Backscattered Ultraviolet (SBUV) satellite data (1998 to 2015). These results are consistent with our negative to near-zero ERA5 trend in the lower stratosphere and the unweighted full lidar trend, but they disagree with our corrected lidar trends. These conflicting results might be due to unconsidered inhomogeneities in lower-stratospheric lidar data, but they also indicate that the chosen uncertainties for weighted trends may not be appropriate in the lower stratosphere where interannual variability is high. To further clarify lower-stratospheric lidar trends, one could investigate absolute trends from ozone number densities instead of volume mixing ratios and use additional data from ozonesondes. Generally, additional analyses are required to derive lower-stratospheric ozone trends with confidence. Indeed, whether lower-stratospheric ozone concentrations increase or continue to decrease is an ongoing discussion [58–61].

Our study concentrates on lidar and MWR datasets with high spatial or temporal resolutions in the middle stratosphere and on ERA5 reanalysis data at Lauder. Other ozone measurements at Lauder from ozonesondes, FTIR, and Umkehr are available but provide data with limited altitude range (ozonesonde) or with smaller vertical resolution (Umkehr and FTIR). Nevertheless, comparing lidar and MWR data with these datasets might be useful to further identify possible data inhomogeneities. The same is true for additional comparision with merged satellite datasets. In future studies, the use of corrected trend estimates could be further improved by automatizing the detection of inhomogeneities. This could be achieved, for example, by defining thresholds of differences when comparing multiple datasets [49] or by constructing an ozone composite using a comprehensive Bayesian approach as presented by Ball et al. [62].

## 6. Conclusions

We presented stratospheric ozone time series from a microwave radiometer (MWR), a lidar, Aura MLS satellite data, and ERA5 reanalysis data from Lauder, New Zealand. We investigated and compared the time series to verify whether they can be used for trend estimation. We then presented ozone trend estimates using two regression models with weighted and unweighted regression.

The lidar and MWR data at Lauder agree well and were judged to be suitable for trend estimation. Nevertheless, accounting for small instrumental changes in the lidar data might improve the trend estimates. In contrast, the ERA5 data show some biases and have to be corrected when estimating trends. The LOTUS and the KIT regression methods have both been tested to obtain best estimates of the true ozone trend. Considering data uncertainties

by using weighted regression changes trend estimates, but further investigations might be required for the use of the weighted LOTUS regression model. We identified data inhomogeneities and recommend considering them in the trend estimation to obtain optimal trend estimates. The ozone data at Lauder report positive ozone trends throughout the middle and upper stratosphere between 0% and 5%, which confirms ozone recovery at these altitudes. In the lower stratosphere, trends differ when uncertainties are included in the weighted regression, suggesting that further analyses are required to derive robust corrected trends in the lower stratosphere.

In summary, our study compares ozone datasets at the Lauder site and shows that they are generally suitable for trend estimation. The agreement of observed ozone anomalies from the four datasets is remarkable and indicates that lidar, MWR, Aura MLS, and ERA5 data at Lauder are highly reliable. However, we also show that some inhomogeneities in the data influence the trend estimates and that differences in how data uncertainties are treated will affect the calculated trend. The results of our study are useful for other ozone trend studies that aim to understand differences in stratospheric ozone trend estimates.

**Author Contributions:** Conceptualization, L.B. and K.H.; Data curation, I.B., G.N., R.Q. and D.S.; Formal analysis, L.B.; Resources, I.B., G.N., R.Q. and D.S.; Supervision, K.H.; Visualization, L.B.; Writing—original draft, L.B.; Writing—review and editing, L.B., I.B., G.N., R.Q., D.S. and K.H. All authors have read and agreed to the published version of the manuscript.

**Funding:** This research was funded by the Swiss National Science Foundation, grant number 200021-165516.

**Data Availability Statement:** The data presented in this study are openly available. Lidar data are available at https://doi.org/10.21336/gen.0x48-sm13 [39] and microwave radiometer data at https://doi.org/10.21336/gen.bpqv-7z42 [30]. Both datasets are also available at http://www.ndaccdemo.org/ [25]. Aura MLS data are available at https://doi.org/10.5067/Aura/MLS/DATA2017 [27]. ERA5 data can be downloaded from https://doi.org/10.24381/cds.bd0915c6 [28]. The LOTUS regression can be obtained from https://arg.usask.ca/docs/LOTUS_regression/, and the KIT model is available on request. The 10.7 cm solar radio flux is provided as a service by the National Research Council of Canada and distributed in partnership with Natural Resources Canada.

**Acknowledgments:** We thank the LOTUS group for the LOTUS regression model, Thomas von Clarmann for providing the KIT trend model, Gunter Stober for the seasonal fitting programme, and Simon Milligan for the language support. We also thank Sophie Godin-Beekmann and Daan Hubert from the LOTUS group for the motivation and ideas for the study. This study contains modified Copernicus Atmosphere Monitoring Service information (2020). Neither the European Commission nor ECMWF is responsible for any use that may be made of the Copernicus information or data it contains.

**Conflicts of Interest:** The authors declare no conflict of interest. The funders had no role in the design of the study; in the collection, analyses, or interpretation of data; in the writing of the manuscript, or in the decision to publish the results.

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
