# Peer review of "Validation and Trend Analysis of Stratospheric Ozone Data from Ground-Based Observations at Lauder, New Zealand"

_remotesensing, doi:10.3390/rs13010109_

Round 1
Reviewer 1 Report
Comments on “Validation and Trend Analysis of Stratospheric Ozone Data from Ground-Based Observations at Lauder, New Zealand” by Bernet et al.
This study investigated and compared the time series of stratospheric ozone time series of microwave radiometer (MWR) and lidar from Lauder, New Zealand with that of ERA5 reanalysis data to verify whether the ground-based ozone data at Lauder, New Zealand can be used for trend estimation. The ozone trend calculated by two regression models with weighted and unweighted regression, and considered that the lidar and MWR data at Lauder are suitable for trend estimation. This is a manuscript on the analysis and evaluation of observations and there are no complex mechanisms involved. The organization of the manuscript is also good. However, the manuscript has some of the following problems, in particular, some key information is not included and the authors need to make this clearer in the manuscript.
- In the introduction, the authors should briefly explain the importance of ozone research. For example, to protect organisms, to influence climate, etc.
- Why is ozone at mid-latitudes in the southern hemisphere important? MLS and Lauder are only a few years apart in length, MLS is more comprehensive detection data, why is Lauder invaluable? What is the advantage of Lauder data over MLS data?
- The 2 regression models should be described in detail in the Methods section. Although there are citations, it is not possible for the reader to consult the citations for every piece of information.
- How the predictors, e.g., qbo, enso, are obtained in the trend model. What index is used by enso? What level of wind is used by qbo?
- How the weight is performed?
- Why is the MWR data only up to 2016? Are there any follow-up observations?
- There are clearly more lines than the four colors in the upper right corner in Figure 1, please explain.
- As can be seen in Figure 1, MLS data at Lauder has been available since 2004. Trends from 2004 should be analyzed so that it makes more sense to compare MWR and lidar trends with MLS trends than with ERA5 information alone.
- Why Figure 2 has no comparison with MLS?
- The difference between ERA5 and observations is mainly in the range of 40-50 km in Figure 2. Why is the difference in their trends mainly in 30 km in Figures 5 and 6?
- Figure 5a and 6a, why is the difference between lidar full and lidar fit so large under 20km? The author only mentions the phenomenon and does not explain why.
- What is the direct linear trend of the MWR and lidar data without regression analysis?
Reviewer 2 Report
Comments on Validation and Trend Analysis of Stratospheric Ozone Data from Ground-Based Observations at Lauder, New Zealand
This article presents stratospheric ozone lidar trends above Lauder using lidar, MLS, and ERA5 data. The data is regressed using LOTUS and a KIT method and corrections and weightings are applied to the lidar data in an effort to correct for sampling errors. I recommend this study for publication after minor revisions.
General Comments:
This is a very well written paper. Determining ozone trends (particularly in the lower stratosphere) using multiple instruments is an important and challenging task. It is also very useful to publish comparisons of LOTUS and KIT using high resolution lidar data. Most of my comments relate to requesting more details in the text, making figures a bit easier for the reader, and I have some concerns with the fitting and weighting of the lidar data. If there is one major change that I would recommend it is to spend a little more time on the lower stratospheric trends - these are still under debate and you could contribute something very useful to the discussion.
Specific comments:
P2L38-39: “Particularly, we investigate whether the datasets agree.” This statement is vague. Please consider adding some quantitative limits for ‘agreement’
P2L39: “measurement failures” is too harsh. Consider ‘measurement biases’
P2L42-44: Please give another sentence or two describing the differences between KIT and LOTUS. What are the expected differences and what do those differences imply? All readers may not be intimately familiar with both techniques.
P3L96-97: How many individual MLS 'profiles' occur within a 2 degree latitude band? The satellite overpass occurs in a very short period of time. At mid-latitudes there might only be 7 or 8 profiles in a 10 degree latitude band during a single overpass. And there is some degree of sampling uncertainty mixed in with the geophysical variability.
Have you conducted a quick sensitivity study?
P3L106-107: Please provide more information about how you remove seasonal and decadal trends. Fitting parameters, proxy variables, assumptions about trends etc.
P3L113: change to ‘ERA5 deviates from MWR’
P3L114: change to ‘ERA5 deviates from lidar’ - The high resolution local measurement is more authoritative than the global reanalysis
P4 Figure 1: Please consider changing the lines corresponding to the uncertainty estimates. Perhaps light dotted lines? They currently make it difficult to see the other lines. Perhaps describe the uncertainty in the text as well.
P4L126: change “also” to ‘is more likely’
P4L133: Is the change in the lidar-ERA5 comparison after the introduction of MLS in 2004 an example of a ‘block’ discussed later in the KIT algorithm? If so please let the reader know here.
P5L145: “The latter is, however, negligible in our study period.” Please provide a citation
P5L145-146: Is the inflection point 'sharp' or is there some transition function? Please provide more information.
P5L162-165: Please provide an equation for this process. I get confused reading math described in words.
P6L173-180: Please relate this back to your datasets. What does a typical block of correlated data look like? How do you determine the number of blocks?
P6L183-191: So you manually define the blocks based on inspection of the data? Please consider clarifying the paragraph starting at line 173. I have a basic idea of what you’re doing but I don’t fully understand how it has been implemented.
P7 Figure 4: This figure is a bit small and the colours are not helpful. Please consider remaking this figure for easier interpretation by the reader.
P8 Figure 5: Please consider adding some grid lines for the reader. The shaded uncertainties are a bit confusing please consider modifying.
P8L229-230: The weighted and fitted trends in the lower stratosphere look like a real problem to me. The uncertainties are massive and the trends greater than 15% are unrealistic. The unmodified lidar data in Fig. 5a seems more convincing. Have you tried weighting and fitting the data below 20 km separately?
Please add more information and discussion about the lower stratospheric portion of the figure. It immediately draws your attention and there is very little explanation or interpretation given.
P9 Figure 6: caption typo “lidat” change to ‘lidar’
P9 Figure 6: same comment as Fig 5. Please consider gridlines
P9 Figure 6: Really nice result!!! Same comment as for the LOTUS. Needs more explanation and discussion of lower stratospheric lidar. For future work it may be interesting to play with the regression parameters, add in some extra proxies, modify the fitting and weighting parameters etc. It would be interesting to see what it takes to bring the lidar trends below 20 km back to reasonable values.
P9L268-271: Yes definitely. The artifacts introduced into the lidar trend profile are disturbing and indicate that the weighting functions are unduly influencing the profile.
P9L278: “mostly” is vague. Please add specifics of where changes do and do not lie within uncertainties.
P10L295: These trends got larger after corrections. Discuss possible fixes in future work.

Round 2
Reviewer 1 Report
I appreciate that the authors have seriously dealt with my concerns. And I think the revised version of the manuscript is an improvement and I have no further scientific questions. The following two references are worth reviewing in the introduction. This manuscript will be accepted for publication after the minor revision.
References:
Calvo N, Polvani LMand Solomon S 2015 Onthe surface impact of Arctic stratospheric ozone extremes Environ. Res. Lett. 10,094003.
Xie, F., J. Li, W. Tian, Q. Fu, F-F. Jin, Y. Hu, J. Zhang, W. Wang, C. Sun, J. Feng, Y. Yang and R. Ding, 2016: A connection from Arctic stratospheric ozone to El Niño-Southern oscillation. Environ. Res. Lett., 11, 124026.
Author Response
Dear reviewer,
thank you for the positive feedback and the final suggestion. We added the two suggested references and some additional references in the introduction as exemplary literature for the chemical, radiative, and dynamic importance of ozone.
You find attached the manuscript with marked changes.
Thank you for your effort and best regards,
Leonie Bernet
